# On the Choice of Variable for Quantization of Conformal GR

**A. B. Arbuzov** [1,2,*] and **A. A. Nikitenko** [1]

1    Bogolyubov Laboratory of Theoretical Physics, Joint Institute for Nuclear Research, Joliot-Curie St. 6, Dubna 141980, Russia; nikitenko@theor.jinr.ru
2    Department of Fundamental Problems of Microworld Physics, Dubna State University, Dubna 141982, Russia
*    Correspondence: arbuzov@theor.jinr.ru

**Abstract:** The possibility of using spin connection components as basic quantization variables of a conformal version of general relativity is studied. The considered model contains gravitational degrees of freedom and a scalar dilaton field. The standard tetrad formalism is applied. Properties of spin connections in this model are analyzed. Secondary quantization of the chosen variables is performed. The gravitational part of the model action turns out to be quadratic with respect to the spin connections. So at the quantum level, the model looks trivial, i.e., without quantum self-interactions. Meanwhile the correspondence to general relativity is preserved at the classical level.

**Keywords:** general relativity; tetrad formalism; conformal invariance; quantum gravity

## 1. Introduction

It is widely known that in the quantization of general relativity (GR), we face the problem of non-renormalizability of the theory if we choose the space-time metric as the basic variables. Non-renormalizability reveals itself starting from the 1-loop order in perturbation theory. Therefore, with this approach, it is possible to construct a quantum theory of gravity only in the one-loop approximation. To solve this problem, various approaches have been proposed. The most popular of these are the effective field theory approach (extending the Lagrangian of general relativity by terms quadratic in curvature and additional scalar fields), string theory, loop quantum gravity, and the method of functional integration.

The approach, which is called effective field theory, is based on the idea that the Lagrangian of general relativity can be considered as a low-energy approximation of a more general theory, which, as a rule, contains terms quadratic in curvature, such as the curvature square and the Gauss–Bonnet term, and often some interaction with a scalar field. Such theories can solve the problem of renormalization, but as a result, ghost and other instabilities can arise and, accordingly, there emerge problems with unitarity. In addition, this approach by default does not address the most important issues of the evolution of topology and the causal structure of space-time. The global hyperbolicity is postulated by default from the very beginning, and the theory of gravity is then treated in the standard way, by analogy with non-gravitational field theories.

In this work, we study the conformal modification of the general theory of relativity (conformal GR) and analyze the possibility of its quantization in variables that were proposed to be used in [1,2]. The main idea of the conformal modification of GR is that instead of the invariance with respect to the group of general covariant transformations $GL(4)$, the invariance with respect to a broader group of transformations is considered [3]. In this case, the action of the original GR undergoes a conformal (Weyl) transformation, as a result of which there appear factors which depend on the dilaton field $D$. In particular, for the interval, one obtains

$$g_{\mu\nu}dx^\mu \otimes dx^\nu = e^{-2D}\, \widetilde{g}_{\mu\nu}\, d\chi^\mu \otimes d\chi^\nu, \tag{1}$$

where $\widetilde{g}_{\mu\nu}$ is the conformal metric. It is assumed that the actually observable quantities are not the standard quantities of GR, but quantities that are conformally equivalent to them.

In the standard GR, Einstein's equations are obtained as the result of variations of the Einstein–Hilbert action

$$S_{\text{Hilbert}} = \int d^4 x \sqrt{-g} \left[ \frac{M_P^2}{16\pi}(R - 2\Lambda) + L_{\text{matter}}(g_{\mu\nu}) \right]. \tag{2}$$

When considering conformal general relativity (CGR), the Einstein–Hilbert action is transformed into the action [4,5]

$$S_{\text{CGR}} = \int d^4 \chi \sqrt{-\widetilde{g}} \left[ \frac{\widetilde{M}_P^2}{16\pi}\left(\widetilde{R} - 2\widetilde{\Lambda}\right) + \frac{3\widetilde{M}_P^2}{8\pi}\left(\widetilde{g}^{\mu\nu}\nabla_\mu D \nabla_\nu D\right) + L_{\text{matter}}(\widetilde{g}_{\mu\nu}) \right]. \tag{3}$$

Here, $\Lambda$ is the cosmological constant, $\widetilde{\Lambda} = e^{-2D}\Lambda$ is the conformal cosmological constant, $M_P$ is the Planck mass, and $\widetilde{M}_P = M_P e^{-D}$ is the conformal Planck mass. Note that action (3) is not equivalent to the Einstein–Hilbert action due to the fact that it is invariant under conformal transformations, while the Einstein–Hilbert action is not. Thus, conformal general relativity is not equivalent to standard GR, i.e., it is actually a modified theory of gravity, as discussed above. Therefore, we will call action (3) one of conformal GR and denote it $S_{\text{CGR}}$. It is important to note that this model is not a special case of the well-known scalar-tensor Brans–Dicke gravity, since it corresponds to the choice of the Dicke constant equal to $-3/2$, which leads to a singularity [4].

The phenomenological properties of the conformal modification of GR were considered in [6–8], in which much attention was paid to the study of data from type Ia supernovae. Based on the results of calculations and subsequent analysis, the authors come to the conclusion that the main contribution to cosmological density comes from matter having a rigid, rather than a vacuum (dark energy), equation of state. This makes it possible to explain the observed accelerated expansion of the universe without using the cosmological constant.

Let us note in advance that below we use four types of indices as follows: four-dimensional coordinate indices without brackets will be denoted by Greek letters and run through the values $0\ldots 3$; three-dimensional coordinate indices without brackets from the middle of the Latin alphabet $i, j \ldots$ denote three-dimensional spatial indices and take values $1, 2, 3$; tetrad indices are enclosed in parentheses. Tetrad indices, like coordinate ones, can take the values $0\ldots 3$ (four-dimensional space-time) or $1, 2, 3$ (three-dimensional spatial).

The article is structured as follows: First, the tetrad formalism and spin connections will be considered as applied to conformal general relativity, and it will be shown that the formalism in question is quite general and applicable to many modified theories of gravity. Then, the dilaton degree of freedom will be extracted from the standard GR metric. Next, the metric of a conformal, plane gravitational wave will be considered and the possibility of its quantization in a standard way will be analyzed.

## 2. Spin Connections

The method that we develop in this work relies heavily on the tetrad formalism and the concept of spin connectivity. In particular, the variables $\omega^R_{(a)(b),(c)}$, which we propose below to consider as the basic variables of quantum gravity, are directly related to the spin connection. Therefore, the current section is devoted to a discussion of the concepts of tetrad formalism and spin connectivity as applied to conformal general relativity.

In the classical general relativity, space-time $(\mathbb{M}, \mathbf{g})$ is described using a model of a smooth four-dimensional Hausdorff manifold $\mathbb{M}$ endowed with a semi-Riemannian metric $g_{\mu\nu}$. Affine connection, which defines the rules for parallel transfer of tensors from one tangent space to another, is considered symmetric and consistent with the metric. In this situation, the affine connection is completely described by a metric and it is represented in

components by Christoffel symbols $\Gamma^\alpha_{\beta\gamma}$. Unfortunately, covariant differentiation of spinor fields in the standard coordinate representation turns out to be a poorly defined operation. The solution to this problem is carried out by passing to the orthonormal frame [9]. General relativity can be formulated in terms of tetrads; the resulting method is called the tetrad formalism [10,11] or the moving frame method. Using this method, one can introduce the concept of a spin connection defined on a frame bundle and correctly define the covariant differentiation procedure for spinor fields. A moving frame (tetrad) given on a manifold if a basis of vectors $e_{(a)}$ is chosen in the tangent space to each point of the manifold [12].

Tetrad and co-tetrad can be expanded in a coordinate basis in the tangent and cotangent spaces, respectively. The expansion of tetrad vectors with respect to a basis in the tangent space has the form

$$e_{(a)} = e^\alpha{}_{(a)}\partial_\alpha, \tag{4}$$

The decomposition of a co-tetrad in the basis of 1-forms (covectors) from the cotangent space has the form

$$e^{(a)} = e_\alpha{}^{(a)}dx^\alpha. \tag{5}$$

It is assumed that the matrices $e^\alpha{}_{(a)}$ and $e_\alpha{}^{(a)}$ are non-degenerate and are sufficiently smooth functions of points of the manifold. The following conditions on co-tetrads are imposed:

$$e^\alpha{}_{(a)}e_\alpha{}^{(b)} = \delta^{(b)}_{(a)}, \tag{6}$$

$$e^\alpha{}_{(a)}e_\beta{}^{(a)} = \delta^\alpha_\beta. \tag{7}$$

Let us also determine quantities $e_{\nu(a)}$ using the formula

$$e_{\nu(a)} = \eta_{(a)(d)}e_\nu{}^{(d)}, \tag{8}$$

where $\eta_{(a)(d)}$ is the Minkowski metric. Since $\eta_{(a)(d)}$ are constants and do not depend on the point of the manifold, then $\eta_{(a)(d)}$ can be freely brought in and out of the derivative.

Unlike coordinate basis vector fields, tetrad vector fields, generally speaking, do not commute with each other. This is due, in particular, to the fact that a pair of arbitrary vector fields on a manifold, generally speaking, do not commute with each other. Geometrically, this means that a small parallelogram constructed using two arbitrarily chosen vector fields will, in the general case, be open. The amount of "openness" in the leading order is precisely expressed by the commutator of vector fields $\left[e_{(a)}, e_{(b)}\right]$. Let us decompose the commutator of a pair of tetrad vector fields into tetrad vectors

$$\left[e_{(a)}, e_{(b)}\right] = c_{(a)(b),}{}^{(c)}e_{(c)}. \tag{9}$$

The coefficients $c_{(a)(b),}{}^{(c)}$ are called nonholonomic coefficients. An explicit expression for the nonholomony coefficients in terms of the tetrad components has the following form:

$$c_{(a)(b),}{}^{(c)} = \left(e^\alpha{}_{(a)}\partial_\alpha e^\beta{}_{(b)} - e^\alpha{}_{(b)}\partial_\alpha e^\beta{}_{(a)}\right)e_\beta{}^{(c)}. \tag{10}$$

The connection on the frame bundle is related to the affine connection, which is given on the tangent bundle by the formula

$$\nabla_\alpha e_\beta{}^{(a)} = \partial_\alpha e_\beta{}^{(a)} - \Gamma_{\alpha\beta}{}^\gamma e_\gamma{}^{(a)} + e_\beta{}^{(b)}\omega_{\alpha(b),}{}^{(a)} = 0. \tag{11}$$

Here, $\omega_{\alpha(a),}{}^{(b)}$ are so-called spin connection components. We are interested in the case when the connection is metric. That is, when the torsion tensor and the nonmetricity tensor vanish: $T_{(a)(b),(c)} = 0$ and $Q_{(a),(b)(c)} = 0$. Note that the torsion tensor is skew-symmetric in the first two indices, and the nonmetricity tensor is symmetric in the last two indices. In this regard, these indices are separated by commas. In this case, the components of the spin connection can be expressed through the nonholonomic coefficients by the formula [12]

$$\omega_{(a),(b)(c)} = \frac{1}{2}\left(c_{(a)(b),(c)} - c_{(b)(c),(a)} + c_{(c)(a),(b)}\right), \tag{12}$$

where the notation $c_{(a)(b),(c)} := \eta_{(c)(d)} c_{(a)(b),}{}^{(d)}$ is used.

The components of the curvature tensor in a nonholonomic basis will have the form

$$
\begin{aligned}
R_{(a)(b),(c)}{}^{(d)} &= \partial_{(a)}\omega_{(b)(c),}{}^{(d)} - \partial_{(b)}\omega_{(a)(c),}{}^{(d)} - \omega_{(a)(c),}{}^{(e)}\omega_{(b)(e),}{}^{(d)} \\
&+ \omega_{(b)(c),}{}^{(e)}\omega_{(a)(e),}{}^{(d)} - c_{(a)(b),}{}^{(e)}\omega_{(e)(c),}{}^{(d)},
\end{aligned}
\tag{13}
$$

where $\omega_{(a)(b),}{}^{(c)} := e^\alpha{}_{(a)}\omega_{\alpha(b),}{}^{(c)}$, $\partial_{(a)} := e^\alpha{}_{(a)}\partial_\alpha$, or if the two indices of the two-dimensional direction in which the curvature is defined remain coordinate

$$R_{\mu\nu,(c)}{}^{(d)} = \partial_\mu\omega_{\nu(c),}{}^{(d)} - \partial_\nu\omega_{\mu(c),}{}^{(d)} - \omega_{\mu(c),}{}^{(e)}\omega_{\nu(e),}{}^{(d)} + \omega_{\nu(c),}{}^{(e)}\omega_{\mu(e),}{}^{(e)}. \tag{14}$$

In paper [1], the introduction of variables $\omega^R_{(a)(b),(c)}$ was discussed and it was argued that they can only be introduced using a nonlinear representation of the symmetry group, but not within the framework of the classical general relativity. Note that such statements should be treated with caution. Now, we will show that, in fact, it is possible to distinguish $\omega^R_{(a)(b),(c)}$ and $\omega^L_{(a)(b),(c)}$ also in the classical GR. And their introduction does not require either conformal transformations with the release of the dilaton or the use of a nonlinear symmetry realization. It is enough to consider the bundle of frames with the structure group $SO(1,3)$ and use the property that the torsion and nonmetricity tensors are equal to zero. In fact, we omit the index in Formula (10) and substitute into Equation (12); taking into account the indices, we obtain

$$c_{(a)(b),(c)} = \left(e^\alpha{}_{(a)}\partial_\alpha e^\beta{}_{(b)} - e^\alpha{}_{(b)}\partial_\alpha e^\beta{}_{(a)}\right)e_{\beta(c)}, \tag{15}$$

$$c_{(b)(c),(a)} = \left(e^\alpha{}_{(b)}\partial_\alpha e^\beta{}_{(c)} - e^\alpha{}_{(c)}\partial_\alpha e^\beta{}_{(b)}\right)e_{\beta(a)}, \tag{16}$$

$$c_{(c)(a),(b)} = \left(e^\alpha{}_{(c)}\partial_\alpha e^\beta{}_{(a)} - e^\alpha{}_{(a)}\partial_\alpha e^\beta{}_{(c)}\right)e_{\beta(b)}, \tag{17}$$

and the expression for the spin connection components

$$
\begin{aligned}
\omega_{(a),(b)(c)} &= \frac{1}{2}\left(e^\alpha{}_{(a)}\partial_\alpha e^\beta{}_{(b)} - e^\alpha{}_{(b)}\partial_\alpha e^\beta{}_{(a)}\right)e_{\beta(c)} - \frac{1}{2}\left(e^\alpha{}_{(b)}\partial_\alpha e^\beta{}_{(c)} - e^\alpha{}_{(c)}\partial_\alpha e^\beta{}_{(b)}\right)e_{\beta(a)} \\
&+ \frac{1}{2}\left(e^\alpha{}_{(c)}\partial_\alpha e^\beta{}_{(a)} - e^\alpha{}_{(a)}\partial_\alpha e^\beta{}_c\right)e_{\beta(b)},
\end{aligned}
\tag{18}
$$

By rearranging the terms in the expression (18), we can distinguish $\omega^L$ and $\omega^R$ in it

$$
\begin{aligned}
\omega_{(a),(b)(c)} &= \frac{1}{2}e^\alpha{}_{(a)}\left(e^\beta{}_{(c)}\partial_\alpha e_{\beta(b)} - e^\beta{}_{(b)}\partial_\alpha e_{\beta(c)}\right) + \frac{1}{2}e^\alpha{}_{(b)}\left(e^\beta{}_{(a)}\partial_\alpha e_{\beta(c)} + e^\beta{}_{(c)}\partial_\alpha e_{\beta(a)}\right) \\
&- \frac{1}{2}e^\alpha{}_{(c)}\left(e^\beta{}_{(b)}\partial_\alpha e_{\beta(a)} + e^\beta{}_{(a)}\partial_\alpha e_{\beta(b)}\right) := \omega^L_{(c)(b),(a)} + \omega^R_{(a)(c),(b)} - \omega^R_{(b)(a),(c)}.
\end{aligned}
\tag{19}
$$

To derive the Formula, (19) was also used $e_{\beta(c)}\partial_\alpha e^\beta{}_{(b)} = -e^\beta{}_{(b)}\partial_\alpha e_{\beta(c)}$, which follows from the fact that $\eta_{(c)(b)} = e_{\beta(c)}e^\beta{}_{(b)}$. The Formula (19) differs in sign from those presented in [1,2], but this does not in any way affect the results presented below.

$$\omega^L_{(c)(b),(a)} = \frac{1}{2}e^\alpha{}_{(a)}\left(e^\beta{}_{(c)}\partial_\alpha e_{\beta(b)} - e^\beta{}_{(b)}\partial_\alpha e_{\beta(c)}\right)$$
$$= \frac{1}{2}\left(e^\beta{}_{(c)}\left(e^\alpha{}_{(a)}\partial_\alpha e_{\beta(b)}\right) - e^\beta{}_{(b)}\left(e^\alpha{}_{(a)}\partial_\alpha e_{\beta(c)}\right)\right), \tag{20}$$

$$\omega^R_{(a)(c),(b)} = \frac{1}{2}e^\alpha{}_{(b)}\left(e^\beta{}_{(a)}\partial_\alpha e_{\beta(c)} + e^\beta{}_{(c)}\partial_\alpha e_{\beta(a)}\right)$$
$$= \frac{1}{2}\left(e^\beta{}_{(a)}\left(e^\alpha{}_{(b)}\partial_\alpha e_{\beta(c)}\right) + e^\beta{}_{(c)}\left(e^\alpha{}_{(b)}\partial_\alpha e_{\beta(a)}\right)\right), \tag{21}$$

$$\omega^R_{(b)(a),(c)} = \frac{1}{2}e^\alpha{}_{(c)}\left(e^\beta{}_{(b)}\partial_\alpha e_{\beta(a)} + e^\beta{}_{(a)}\partial_\alpha e_{\beta(b)}\right)$$
$$= \frac{1}{2}\left(e^\beta{}_{(b)}\left(e^\alpha{}_{(c)}\partial_\alpha e_{\beta(a)}\right) + e^\beta{}_{(a)}\left(e^\alpha{}_{(c)}dx^{(c)}\partial_\alpha e_{\beta(b)}\right)\right), \tag{22}$$

In [13], up to the index notation, $\omega^L_{(a)(c),(\alpha)}$ and $\omega^R_{(a)(c),(\alpha)}$ are expressed as

$$\omega^L_{(a)(c),\alpha}dx^\alpha = \frac{1}{2}\left(e^\beta{}_{(c)}de_{\beta(b)} - e^\beta{}_{(b)}de_{\beta(c)}\right), \tag{23}$$

$$\omega^R_{(a)(c),\alpha}dx^\alpha = \frac{1}{2}\left(e^\beta{}_{(c)}de_{\beta(b)} + e^\beta{}_{(b)}de_{\beta(c)}\right). \tag{24}$$

Note that in deriving these formulas, we never used either conformal symmetry or the nonlinear representation of the extended symmetry group [3]. However, the introduction of $\omega^R_{(a)(b),(c)}$ and $\omega^L_{(a)(b),(c)}$ by means of the nonlinear symmetry group, as described in [1–3], still looks more natural from a physical point of view. This happens through the establishment of a connection $\omega^R_{(a)(b),(c)}$, $\omega^L_{(a)(b),(c)}$ with Goldstone-type fields. For more information about this relationship, see the review article [14].

Note further that when $\omega^R_{(b)(a),(c)}$ and $\omega^L_{(a)(b),(c)}$ are mentioned in the text, we omit the indices where this does not lead to loss of meaning and denote them simply as $\omega^R$ and $\omega^L$, respectively.

It should be noted here that in the general case, the components of the spin connection are expressed through the nonholomonicity coefficients, the metric tensor, the torsion tensor, and the nonmetricity tensor. A spin connection on a tetrad bundle is generally associated with the group $GL(4)$ corresponding to the tetrad rotations. In the general theory of relativity, in the case of a coordinate description, an affine connection is used, which is considered symmetric (zero torsion tensor) and consistent with the metric (zero nonmetricity tensor). When reformulating GR in terms of the tetrad formalism, instead of the affine connection, the spin connection is used, which is considered metric (both nonmetric and torsion tensors are zero) to be consistent with the standard coordinate formulation of GR. The components of the metric spin connection will be expressed only in terms of nonholonomic coefficients. In this case, the reduction of the group $GL(4)$ to the group $SO(1,3)$ follows immediately from the equality of the nonmetricity tensor to zero. This result is essentially a well-known fact (see, for example, monograph [12]). Here, we note that, just as in the case of the tetrad formulation of standard general relativity, we consider the metric spin connection. That is, in the tetrad formulation of conformal GR, the same spin connection is used as in the tetrad formulation of standard GR. It follows from this that considering the spin connection with respect to the $SO(1,3)$ group does not change the essence of the theory, at least not at the classical level.

It is also worth noting here that quantization of the $GL(4)$ theory should be different from quantization of the $SO(1,3)$ theory. Our work is devoted to the consideration of a special set of variables for the quantization of conformal general relativity associated with spin connection and tetrad formalism. That is, we are studying one of many possible approaches. The question of what differences should exist when quantizing a theory with a symmetry group other than $GL(4)$ is an interesting one. A meaningful study of such a difference is by no means trivial and is the topic of a separate work. It is known that the use of tetrads and spin connections helps to construct a quantum theory of spinor fields in a curved space-time. In this regard, in this work, we consider that it is worth considering a similar approach to gravity.

Let us show that the derivatives of the metric tensor cannot depend on the components of $\omega^L$ and, thus, $\omega^L$ cannot play the role of dynamic variables. The coordinate components of the metric tensor are related to the coordinate components of the tetrads by the formula

$$g_{\mu\nu} = e_\mu{}^{(a)} e_{\nu(a)}. \tag{25}$$

$$dg_{\mu\nu} = dx^\alpha \partial_\alpha g_{\mu\nu} = d\left(e_\mu{}^{(a)} e_{\nu(a)}\right) = d\left(e_\mu{}^{(a)}\right) e_{\nu(a)} + e_\mu{}^{(a)} d\left(e_{\nu(a)}\right), \tag{26}$$

Using the Formulas (23) and (24), we have

$$de_\mu{}^{(a)} = e_\mu{}^{(b)} \left(\omega^R{}_{(b)}{}^{(a)},(dx^\alpha) + \omega^L{}_{(b)}{}^{(a)},(dx^\alpha)\right), \tag{27}$$

$$de_{\nu(a)} = e_\nu{}^{(b)} \left(\omega^R{}_{(b)(a)},(dx^\alpha) + \omega^L{}_{(b)(a)},(dx^\alpha)\right), \tag{28}$$

Substituting (27) and (28) into (26), we obtain the following expression for the total differential of the metric tensor:

$$
\begin{aligned}
dg_{\mu\nu} &= dx^\alpha \partial_\alpha g_{\mu\nu} = d\left(e_\mu{}^{(a)} e_{\nu(a)}\right) = d\left(e_\mu{}^{(a)}\right) e_{\nu(a)} + \left(e_\mu{}^{(a)} d\left(e_{\nu(a)}\right)\right) \\
&= e_{\nu(a)} e_\mu{}^{(b)} \left(\omega^R{}_{(b)}{}^{(a)},(dx^\alpha) + \omega^L{}_{(b)}{}^{(a)},(dx^\alpha)\right) + e_\mu{}^{(a)} e_\nu{}^{(b)} \left(\omega^R{}_{(b)(a)},(dx^\alpha) + \omega^L{}_{(b)(a)},(dx^\alpha)\right).
\end{aligned}
\tag{29}
$$

Omitting the index $(a)$ in the expression $\left(\omega^R{}_{(b)(a)},(dx^\alpha) + \omega^L{}_{(b)(a)},(dx^\alpha)\right)$, and at the same time raising $(a)$ to $e_{\nu(a)}$, we take out the common factor and obtain from (29) the following expression:

$$
\begin{aligned}
dg_{\mu\nu} &= dx^\alpha \partial_\alpha g_{\mu\nu} = d\left(e_\mu{}^{(a)} e_{\nu(a)}\right) = d\left(e_\mu{}^{(a)}\right) e_{\nu(a)} + \left(e_\mu{}^{(a)} d\left(e_{\nu(a)}\right)\right) \\
&= e_{\nu(a)} e_\mu{}^{(b)} \left(\omega^R{}_{(b)}{}^{(a)},(dx^\alpha) + \omega^L{}_{(b)}{}^{(a)},(dx^\alpha)\right) + e_\mu{}^{(a)} e_\nu{}^{(b)} \left(\omega^R{}_{(b)(a)},(dx^\alpha) + \omega^L{}_{(b)(a)},(dx^\alpha)\right) \\
&= \left(e_\mu{}^{(b)} e_\nu{}^{(a)} + e_\mu{}^{(a)} e_\nu{}^{(b)}\right) \left(\omega^R{}_{(b)(a)},(dx^\alpha) + \omega^L{}_{(b)(a)},(dx^\alpha)\right) \\
&= \left(e_\mu{}^{(b)} e_\nu{}^{(a)} + e_\mu{}^{(a)} e_\nu{}^{(b)}\right) \omega^R{}_{(b)(a)},(dx^\alpha).
\end{aligned}
\tag{30}
$$

Although in the modification of GR we are considering, where the torsion and non-metricity tensors are equal to zero, the congruences of world lines, just like in classical GR, can have non-zero rotation. When considering dynamic variables, we would like to separate the gravitational properties of the theory from properties that are not directly related to gravity and characterize the congruence of world lines (the reference system). Therefore, the disappearance of $\omega^L$ variables in the equation should be interpreted as the separation of the properties of the reference system (congruence of world lines) associated with rotation from the gravitational properties of the theory.

The standard expression for spin connection components in the absence of torsion and nonmetricity is represented in terms of nonholonomic coefficients (metric connection).

A new result of this Section is the substantiation of the possibility of separating the variables $\omega^R$ and $\omega^L$ from the standard expression for the components of the spin connection, and the possibility of writing the expression for its components in terms of the variables $\omega^R$ and $\omega^L$ in the form in which it appears in earlier works [1,2]. In paper [1], it was argued that to introduce the variables $\omega^R$ and $\omega^L$, the presence of a nonlinear realization of the symmetry group is necessary. In this paper, we show that the variables $\omega^R$ and $\omega^L$ can be introduced in any theory of gravity in which the spin connection is metric.

## 3. ADM Formalism as Applied to Conformal General Relativity

In this section, we will discuss the application of the Arnowitt–Deser–Miesner formalism to conformal general relativity. Here, we will limit ourselves to considering only globally hyperbolic space-times $(\mathbb{M}, \mathbf{g})$. From the condition of global hyperbolicity of space-time and Geroch's splitting theorem [15–17], it follows that $(\mathbb{M}, \mathbf{g})$ is representable as a direct product $\mathbb{M} = \mathbb{R} \times \mathbb{S}$, where $\mathbb{S}$ is a spacelike Cauchy 3-surface (hypersurface). Consequently, for a given space-time, the Arnowitt–Deser–Misner Formalism is valid, which allows us to represent the conformal metric in the form [18]

$$\widetilde{ds}^2 = \widetilde{g}_{ij}(dx^i + N^i dt)(dx^j + N^j dt) - (N_0 dt)^2. \tag{31}$$

As already mentioned in Section 1, the standard metric is related to the conformal metric by the Formula (1). In tetrad representation, the metric will look like

$$
\begin{aligned}
g_{\mu\nu}dx^\mu \otimes dx^\nu &= e^{-2D}\,\widetilde{g}_{\mu\nu}\,d\chi^\mu \otimes d\chi^\nu = e^{-2D}\eta_{(a)(b)}e^{(a)} \otimes e^{(b)} \\
&= e^{-2D}\eta_{(a)(b)}(e_\mu{}^{(a)}dx^\mu)(e_\nu{}^{(b)}dx^\nu). 
\end{aligned}
\tag{32}
$$

The 1-forms $e^{(a)}$ are related to the metric in the Arnowitt–Deser–Miesner representation through the formulas

$$
\begin{cases}
e^{(0)} = N dx^0, \\
e^{(j)} = e_i{}^{(j)}\big[dx^i + N^i dx^0\big].
\end{cases}
\tag{33}
$$

Here, $e^0$, $e^{(j)}$ are a set of basic co-tetrads (1-forms conjugate to tetrad vectors), $e_{(j)i}$. The quantities $N$ and $N^i$ are called the lapse function and the shift vector, respectively. The lapse function can be represented as

$$N(\chi^0, \chi^1, \chi^2, \chi^3) = N_0(\chi^0)\mathcal{N}(\chi^0, \chi^1, \chi^2, \chi^3), \tag{34}$$

having identified in it a part that depends only on time (global) and a part that depends on coordinates (local part) [19]. In (34), $N_0$ is the global part of the lapse function, and $\mathcal{N}$ is the local part of the lapse function. The volume shape then, taking into account (34), can be written as

$$d^4 x \sqrt{-\widetilde{g}} = d\chi^0 N_0\, d^3\chi\sqrt{\gamma}\,\mathcal{N}, \tag{35}$$

where $\gamma$ is the determinant of the spatial metric. This allows us to separately highlight integration over space in action. Action (3) will then have the form

$$S_{\text{Gravitons}} = \int d\chi^0 N_0 \int d^3\chi\sqrt{\gamma}\,\mathcal{N}\,\frac{\widetilde{M}_P^2}{16\pi}\widetilde{R}. \tag{36}$$

Thus, (36) is divided into temporal and spatial parts. We also omitted the $\Lambda$ term, since its presence is not essential for our further analysis.

### 4. Analysis of Variables Proposed for Gravity Quantization

The purpose of this section is to analyze the possibility of constructing a quantum theory of gravitational waves and their interaction with matter within the framework of conformal general relativity if we accept the variables $\omega^R$ as basic variables when quantizing gravity, as was proposed in [1,2].

Quantization of gravity in the general case is known to be an extremely difficult problem. In this regard, following [1] in this work, we will consider the simplest special case of a plane wave and some questions that arise when trying to quantize conformal general relativity. We are interested in the question of quantization of gravitational waves, so we will now focus on the purely gravitational part of the action (3), which we presented in Section 3 as (36), explicitly separating the temporal and spatial parts.

Since in CGR, the quantities that are actually observable in experiment are those that are obtained from standard ones using the Weyl transformations, we need to consider the conformal metric $\widetilde{g}_{\mu\nu}$, which describes the gravitational wave. We will call gravitational waves in conformal general relativity conformal gravitational waves. The conformal metric is obtained from the standard metric using the appropriate conformal transformation (1). Thus, in conformal GR, the main task is to find a conformal transformation that would bring the standard metric to a conformal (actually observable) metric.

In standard general relativity, there are different metrics that can be interpreted as metrics describing gravitational waves. The simplest of them is the nonlinear (strong) plane wave metric, which can be represented in the form [18]

$$g = -d\chi^0 \otimes d\chi^0 + d\chi^3 \otimes d\chi^3 + e^\Sigma \left[ e^\sigma d\chi^1 \otimes d\chi^1 + e^{-\sigma} d\chi^2 \otimes d\chi^2 \right]. \tag{37}$$

Without loss of generality, we can set $e^\Sigma = 1$ and, accordingly, $\Sigma = 0$. It is natural to look for a solution like gravitational waves in CGR, considering metric (37) as an ansatz. Then, taking into account the condition $\Sigma = 0$ imposed above, the metric of the conformal $\widetilde{g}_{\mu\nu}$ gravitational wave (37) can be written in the form

$$\widetilde{g} = -d\chi^0 \otimes d\chi^0 + d\chi^3 \otimes d\chi^3 + e^\sigma d\chi^1 \otimes d\chi^1 + e^{-\sigma} d\chi^2 \otimes d\chi^2. \tag{38}$$

Below, the metric and $\omega^R$ everywhere refer to conformal general relativity; therefore, to simplify the notation, we will further omit the symbol $\sim$ above all conformal quantities. We also need to impose a gauge condition. We choose the so-called Lichnerovich gauge condition [20], which is that the determinant of the conformal three-dimensional metric is $\gamma = 1$. The action for gravity (36) can be written as

$$
\begin{aligned}
S_{\text{Gravitons}} \;=\; & \int d\chi^0 \, d^3\chi \left\{ \frac{1}{2} \left[ \left( \frac{\partial \sigma}{\partial \chi^0} \right)^2 - \left( \frac{\partial \sigma}{\partial \chi^3} \right)^2 \right] \right. \\
& \left. - \; e^{-\Sigma} \left( e^{-\sigma} \frac{\partial^2 \Sigma}{\partial (\chi^1)^2} + e^\sigma \frac{\partial^2 \Sigma}{\partial (\chi^2)^2} \right) \right\}.
\end{aligned}
\tag{39}
$$

We can represent (33) in the equivalent form

$$
\begin{cases}
e_0{}^{(0)} = N, \\
e_i{}^{(0)} = 0, \\
e_0{}^{(a)} = e_i{}^{(a)} N^i.
\end{cases}
\tag{40}
$$

$$e_0{}^{(0)} = 1, \quad e_3{}^{(3)} = 1, \quad e_1{}^{(1)} = e^{\frac{1}{2}\sigma}, \quad e_2{}^{(2)} = e^{-\frac{1}{2}\sigma}, \tag{41}$$

$$e^{0(0)} = -1, \quad e^{3(3)} = 1, \quad e^{1(1)} = e^{-\frac{1}{2}\sigma}, \quad e^{2(2)} = e^{\frac{1}{2}\sigma}. \tag{42}$$

The expression $e_{\sigma(a)}de^{\sigma}{}_{(b)}$ is symmetric in the indices $\mu\nu$, so we can write $\omega^R$ as

$$\omega^R_{(a)(b),\alpha}dx^{\alpha} = e_{\sigma(a)}dx^{\alpha}\frac{\partial e^{\sigma}{}_{(b)}}{\partial x^{\alpha}} = \frac{1}{2}dx^{\alpha}\frac{\partial \sigma}{\partial x^{\alpha}}\left(\delta_{(a)(1)}\delta_{(b)(1)} - \delta_{(a)(2)}\delta_{(b)(2)}\right). \tag{43}$$

Hence, further, as shown in [1,2], the following representation is valid for variables

$$\omega^R_{(a)(b),(c)} = \int \frac{d^3k}{(2\pi)^3}\frac{1}{\sqrt{2\omega_k}}ik_{(c)}\left[\epsilon^R_{(a)(b)}(k)g^+_k e^{ik\cdot x} + \epsilon^R_{(a)(b)}(-k)g^-_k e^{-ik\cdot x}\right], \tag{44}$$

where $g^{\pm}_k$ pretend to be the operators of creation and annihilation of conformal gravitons.

Let us now analyze how gravity interacts with itself and with matter if we accept the variables $\omega^R$ as the fundamental variables of quantum gravity, as proposed in [1,2]. Expression (44) was proposed to be considered as a conformal free plane gravitational wave. First of all, we note that the expression (44) contains momentum $k_{(c)}$. From this alone, we can conclude that this expansion does not describe an ordinary plane wave. Let us first show that in the action (3), there are no terms from which a propagator of a conformal graviton could be obtained. To do this, we will use the expression for the curvature tensor in the nonholonomic basis (13). Making the contraction in the expression (13) using the indices $(b)$, $(d)$, we obtain the Ricci tensor

$$R_{(a)(c)} = R_{(a)(b),(c)}{}^{(b)} = \partial_{(a)}\omega_{(b)(c),}{}^{(b)} - \partial_{(b)}\omega_{(a)(c),}{}^{(b)} - \omega_{(a)(c),}{}^{(e)}\omega_{(b)(e),}{}^{(b)}$$
$$+ \omega_{(b)(c),}{}^{(e)}\omega_{(a)(e),}{}^{(b)} - c_{(a)(b),}{}^{(e)}\omega_{(e)(c),}{}^{(b)}. \tag{45}$$

Now, raising one of the lower indices using the conformal metric $\eta^{(a)(c)}$ and performing convolution, we obtain the following expression for the scalar curvature expressed in terms of the components of the spin connection $\omega_{(b)(c)}{}^{(d)}$:

$$R = \eta^{(a)(c)}R_{(a)(c)} = \eta^{(a)(c)}R_{(a)(b),(c)}{}^{(b)} = \eta^{(a)(c)}\partial_{(a)}\omega_{(b)(c),}{}^{(b)} - \eta^{(a)(c)}\partial_{(b)}\omega_{(a)(c),}{}^{(b)}$$
$$- \eta^{(a)(c)}\omega_{(a)(c),}{}^{(e)}\omega_{(b)(e),}{}^{(b)} + \eta^{(a)(c)}\omega_{(b)(c),}{}^{(e)}\omega_{(a)(e),}{}^{(b)} - \eta^{(a)(c)}c_{(a)(b),}{}^{(e)}\omega_{(e)(c),}{}^{(b)}. \tag{46}$$

Raising the index in the expression for the spin connection (19) through $\omega^R$ and $\omega^L$, up to the designation of the indices, we obtain the formula for its components in the form in which we need it in the expression (45). Since the dynamic variables in this case are $\omega^R$, and not $\omega^L$, we will explicitly write out the formula only for them

$$\omega_{(b),(c)}{}^{(d)} = \eta^{(d)(a)}\left(\omega^L_{(a)(c),(b)} + \omega^R_{(b)(a),(c)} - \omega^R_{(c)(b),(a)}\right). \tag{47}$$

Then, $\omega_{(b),(c)}{}^{(b)}$ is expressed by the formula

$$\omega_{(b),(c)}{}^{(b)} = \eta^{(b)(a)}\left(\omega^L_{(a)(c),(b)} + \omega^R_{(b)(a),(c)} - \omega^R_{(c)(b),(a)}\right), \tag{48}$$

and $\omega_{(a),(c)}{}^{(b)}$ by the formula

$$\omega_{(a),(c)}{}^{(b)} = \eta^{(b)(d)}\left(\omega^L_{(d)(c),(a)} + \omega^R_{(a)(d),(c)} - \omega^R_{(c)(a),(d)}\right), \tag{49}$$

Due to the gauge conditions imposed above $N = 1$, $N_i = 0$ and $\gamma = 1$, only the volume shape components $\sqrt{-g} = 1$ and $\omega^R$ can be contained in the expression calculated above for scalar curvature. Then, by substituting (48) and (49) into (46), taking into account the location of the indices, one can obtain an expression for the derivatives of $\omega^R$, from which it is clear that in the Lagrangian, there are no terms containing the square of derivatives $\omega^R$ from which, after integration by parts, a conformal graviton propagator could arise.

Thus, if we accept $\omega^R$ as the fundamental variables of quantum gravity, then we discover the absence of an analogue of the wave equation for the gravitational variables $\omega^R$ that is familiar to us in quantum field theory.

Let us now find out what the terms of the interaction of gravity with matter look like if we choose $\omega^R$ as the basic variables when quantizing conformal general relativity. First of all, let us note the following fact: In modified theories of gravity, the interaction with matter fields is either minimal or non-minimal. If the interaction is minimal, then the corresponding Lagrangian terms can contain metric components, but not their derivatives. If the interaction under consideration is non-minimal, then the Lagrangian may contain derivatives of the metric components.

Let us look again at action (50). Due to the gauge conditions already imposed above $N = 1$, $N_i = 0$ and $\gamma = 1$, we have $\sqrt{-g} = 1$, and for the action of matter, the Lagrangian will have the form

$$S = \int d^4\chi L_{\text{matter}}(\widetilde{g}_{\mu\nu}). \tag{50}$$

Note that if we consider only the interaction terms in (50), then by virtue of (53), we have

$$dg_{\mu\nu} = dx^\alpha \frac{\partial g_{\mu\nu}}{\partial x^\alpha} = \left(e_\mu{}^{(b)}e_\nu{}^{(a)} + e_\mu{}^{(a)}e_\nu{}^{(b)}\right)\omega^R_{(b)(a),\alpha}dx^\alpha \tag{51}$$

$$= \left(e_\mu{}^{(b)}e_\nu{}^{(a)} + e_\mu{}^{(a)}e_\nu{}^{(b)}\right)\omega^R_{(b)(a),\alpha}dx^\alpha. \tag{52}$$

Accordingly, after passing to tetrad indices using $e_{(c)}{}^\alpha$, we obtain a system of first-order differential equations

$$\frac{\partial g_{\mu\nu}}{\partial x^{(c)}} = \left(e_\mu{}^{(b)}e_\nu{}^{(a)} + e_\mu{}^{(a)}e_\nu{}^{(b)}\right)\omega^R_{(b)(a),(c)}, \tag{53}$$

which must be solved in order to express the components of the metric tensor through the variables $\omega^R$, but in the absence of a propagator for $\omega^R$, it is impossible to construct a standard quantum theory of the interaction of $\omega^R$ with matter fields.

## 5. Conclusions

In this work, some properties of the conformal modification of general relativity related to the spin connections were investigated. In particular, it was shown that the expression for the spin connection components (19) expressed in terms of $\omega^L$ and $\omega^R$ can be obtained through formal transformations of the standard formula for spin connection, which is usually written in terms of nonholonomic coefficients in the form (12). It is demonstrated that conformal symmetry and its nonlinear realization is not mandatory for the introduction of the variables $\omega^{R,L}$ for construction of the spin connection (19). This provides an extension of the approach presented in [1]. Thus, we generalized the expression for the spin connection components (19), written in terms of $\omega^{R,L}$, to a much wider range of theories of gravity. Namely, the components of the spin connection can be represented as (19) in any theory of gravity in which the spin connection is a metric one. Note also that our approach is alternative with respect to the attempts of gravity quantization in terms of tetrads as the basic variables. On the other hand, variables $\omega^{R,L}$ are related to tetrads through the formulae (24) and (23), so the former can be exploited in the general tetrad formalism.

We also performed an analysis of the construction of quantum gravity formulated on the basis of the standard quantum field approach in terms of $\omega^R$ variables in Section 4. It was shown that the gravitational Lagrangian not only lacks interaction terms, as has been noted in [1], but also does not contain terms being quadratic in derivatives of $\omega^R$, of which, after integration by parts, a propagator of the conformal graviton would arise in the terms of variables $\omega^R$. Here, one can note a similarity to quantization of a free Dirac spinor field. One can see that the constructed (strong) gravitational waves do not interact

either with each other or with any matter field. So, the obtained quantum theory looks trivial. Nevertheless, there are still minimal couplings (of the classical type) to the matter field and gravity though the metric tensor. Note that the classical GR is reproduced.

As a result, we come to the conclusion that formulation of quantum gravity in terms of $\omega^R$ as the fundamental quantum variables yields an almost trivial theory, which is, nevertheless, worthy of further investigation. In particular, emission, propagation, and detection of the gravitational waves have to be analyzed. Moreover, there is another issue for further studies, associated with the fact that quantities $\omega^R$ do not form a tensor, and, thus, covariant integration by parts in an arbitrary curved space-time turns out to be an ill-defined operation (the results will depend on the parameterization). In any case, we argue that spin connections are definitely useful for studies of its properties in the tetrad formalism, as in general relativity itself, as well as in its modifications, including the conformal one.

**Author Contributions:** Conceptualization, A.B.A.; methodology, A.B.A., A.A.N.; investigation, A.B.A., A.A.N.; writing—original draft, A.A.N.; supervision, A.B.A. All authors have read and agreed to the published version of the manuscript.

**Funding:** This research received no external funding.

**Data Availability Statement:** Data are contained within the article.

**Conflicts of Interest:** The authors declare no conflict of interest.

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
