# Peer review of "On the Choice of Variable for Quantization of Conformal GR"

_universe, doi:10.3390/universe10070294_

Round 1

Reviewer 1 Report

Comments and Suggestions for Authors

I find the paper to be of low scientific quality. I do not understand how the title of the paper corresponds to the actual content. There are also some statements that I believe are erroneous. 

In more details. (1) The statement in line 66 " This in turn means that 66 conformal GR has three degrees of freedom" is incorrect. This is because S_CGR[tilde{g},D] = S_GR[e^{-D} tilde{g}] and the theory the authors call CGR still has two propagating DOF, as GR. This is because it possesses an additional gauge invariance which renders the seemingly present additional propagating mode an illusion. Deviations from GR only arise when matter is present, but in the absence of matter this theory is equivalent to GR. This theory has been discussed in the literature, in the context of scalar-tensor theories of gravity. 

(2) Content of section 2 is standard, and it is not clear what the authors attempt to establish here. I do not see any new results in this section. Same applies to section 3. 

(3) This appears to be the main section of the paper, at least it correlates with the title. But the discussion proceeds with a series of rather random manipulations. It is not clear how the plane metric (38) is related to the manipulations starting with (45). 

Overall, it is not clear to this referee what this paper is about, and what are results if any. I cannot recommend its publication. 

Comments on the Quality of English Language

English is readable. 

Author Response

Comment 1:

The statement in line 66 " This in turn means that 66 conformal GR has three degrees of freedom" is incorrect. This is because $S_{CGR}[\tilde{g},D] = S_{GR}[e^{-D} \tilde{g}]$ and the theory the authors call CGR still has two propagating DOF, as GR. This is because it possesses an additional gauge invariance which renders the seemingly present additional propagating mode an illusion. Deviations from GR only arise when matter is present, but in the absence of matter this theory is equivalent to GR. This theory has been discussed in the literature, in the context of scalar-tensor theories of gravity.

Reply 1:

We agree with the reviewer that the action of conformal general relativity is obtained from the Einstein-Hilbert action through a conformal transformation and in this sense should be equivalent to the original action at the classical level, preserving the number of independent degrees of freedom. However, the resulting action can also be viewed as an independent expression, which is the initial postulate. Since it contains additional symmetry, we can expect differences at the quantum level in the presence of matter. The article is specifically devoted to discussing the choice of variables for quantizing the theory. Our choice of spin connections as basic variables is motivated by the fact that they are associated with generators of group transformations. 

Changes have been made to the text of the article (the changes are in bold face). The phrase about three degrees of freedom has been removed.

Comment 2:

Content of section 2 is standard, and it is not clear what the authors attempt to establish here. I do not see any new results in this section. Same applies to section 3.

Reply 2:

The standard expression for the components of the spin connection in the absence of torsion and non-metricity is expressed in terms of the nonholonomic coefficients (metric connection). A new result of this Section is the substantiation of the possibility of separating the variables $\omega^{R}$ and $\omega^{L}$ from the standard expression for the components of the spin connection and the possibility of writing the expression for its components in terms of the variables $\omega^{R}$ and $\omega^{L}$ in the form in which it appears in earlier works, see [1],[2]. In ref. [1], it was argued that to introduce the variables $\omega^{R}$ and $\omega^{L}$ the presence of a nonlinear realization of the symmetry group is necessary. In this paper, we show that the variables $\omega^{R}$ and $\omega^{L}$ can be introduced in any theory of gravity in which the spin connection is metric. The key result of this Section is the justification that it is $\omega^{R}$ (and not $\omega^{L}$) that can be considered as dynamic variables. Explanations have been added at the end of the "Spin connections" section.

Comment 3:

We consider $\omega^{R}$ variables as the basic variables of quantum gravity, and then we consider the special case of a free conformal gravitational wave. After obtaining representation (44), we analyze the possibility of quantizing the free gravitational field in the standard way in the variables $\omega^{R}$.

Comment 4:

Overall, it is not clear to this referee what this paper is about, and what are results if any. I cannot recommend its publication.

Reply 4:

Our article is devoted to the development of an alternative approach of gravity quantization. It contains technical details on separation of the proposed quantization variables and justification of the quantization procedure.
The key point of the procedure is the direct analogy with the quantization of a spinor field in a curved space-time, i.e., first we go to the tangent space and only then we perform quantization.
We argue that the constructed quantization scheme requires further verification and suggest some further steps.

Reviewer 2 Report

Comments and Suggestions for Authors

REFEREE REPORT

Title: On the choice of variable for quantization of conformal GR

Authors: A.B. Arbuzov, A.A. Nikitenko

In this manuscript, the authors study conformal general relativity as a viable theory for the gravitational field, analyze its geometric structure from the point of the tetrad formalism, and propose a particular section of the spin connection components as the right variables for a possible quantization of the theory. Although all the calculations seem to be correct, I think that this manuscript needs major revision before being considered for publication. I recommend that the authors consider the following points.

1. In the introduction and also Sec. 2, it is stated and presented as a result of this work that the tetrad formalism can be applied to other modified theories. This is not correct. In fact, the tedrad formalism is an application of the very well-known theory of differential forms that can be applied to any differential manifold, independently of any theory that could be described on this manifold. In fact, the local tetrads and the spin connection are one-forms, which through the exterior derivative determine the curvature two-form. Instead of using the appropriate language of differential forms, the authors use a mixture of tetrad and tensor indices, which are also known as "bastard" objects (see, for instance, the textbook "Gauge fields, knots, and gravity" by Baez & Muniain). I think this point should be clarified in the manuscript.

2. The interesting point of using tetrads with a local Minkowski metric is that the symmetry of the theory is reduced from GL(4) to SO(3,1). The authors should show explicitly that classically this reduction does not affect the theory. This is an exercise on the use of differential forms on the cotangent space and tensors on the tangent space. 

3. The real question is whether the symmetry reduction can affect the quantization of the theory. In fact, the quantization of a GL(4) theory is completely different from the quantization of an SO(3,1) theory. The authors should include a detailed discussion about this issue.

4. The authors propose the spin connection $\omega^R$ as the underlying variable for quantization. Why? Why not $\omega^L$? This theory contains metric and connection as geometric variables (apart from the dilatonic field). Does the choice of $\omega$ implies that the degrees of freedom of the metric should not be quantized?

5. The authors use a particular metric for planar waves to analyze the quantization problem and conclude that the corresponding quantum theory is trivial. This is not correct. They can only conclude that the quantization of planar waves cannot be performed in this way. The point is that fixing a gauge and quantizing is not the same as quantizing and fixing the gauge. The authors should discuss in detail their method of quantization from a physical and mathematical point of view because this is certainly not the standard way of looking at the problem of quantizing gravity.

Comments on the Quality of English Language

Use a speller.

Author Response

We are grateful to the Referee for critical remarks and comments.

Comment 1:

In this manuscript, the authors study conformal general relativity as a viable theory for the gravitational field, analyze its geometric structure from the point of the tetrad formalism, and propose a particular section of the spin connection components as the right variables for a possible quantization of the theory. Although all the calculations seem to be correct, I think that this manuscript needs major revision before being considered for publication. I recommend that the authors consider the following points.

  1. In the introduction and also Sec. 2, it is stated and presented as a result of this work that the tetrad formalism can be applied to other modified theories. This is not correct. In fact, the tedrad formalism is an application of the very well-known theory of differential forms that can be applied to any differential manifold, independently of any theory that could be described on this manifold. In fact, the local tetrads and the spin connection are one-forms, which through the exterior derivative determine the curvature two-form. Instead of using the appropriate language of differential forms, the authors use a mixture of tetrad and tensor indices, which are also known as "bastard" objects (see, for instance, the textbook "Gauge fields, knots, and gravity" by Baez Muniain). I think this point should be clarified in the manuscript.

Reply 1:

We agree that tetrad formalism and the method of differential forms are closely interrelated. In our work, we do use the mixed formalism because it seems convenient for working with variables that we consider to be basic quantization variables. We have tried to describe in detail all the indices and notations used.

Comment 2:

The interesting point of using tetrads with a local Minkowski metric is that the symmetry of the theory is reduced from GL(4) to SO(3,1). The authors should show explicitly that classically this reduction does not affect the theory. This is an exercise on the use of differential forms on the cotangent space and tensors on the tangent space.

Reply 2:

It should be noted here that in the general case, the components of the spin
connection are expressed through: nonholomonicity coefficients, the metric tensor, 
the torsion tensor, and the nonmetricity tensor. A spin connection on a tetrad bundle 
is generally associated with the group GL(4) corresponding to tetrad rotations. In the general
theory of relativity (GR), in the case of a coordinate description, an affine connection is
used, which is considered symmetric (torsion tensor = 0) and consistent with the
metric (nonmetricity tensor = 0). When reformulating GR in terms of the tetrad
formalism, instead of the affine connection, the spin connection is used, which is
considered metric (non-metric tensor = 0, and torsion tensor = 0) to be consistent
with the standard coordinate formulation of GR. The components of the metric
spin connection will be expressed only in terms of nonholonomic coefficients. In
this case, the reduction of the group GL(4) to the group SO(1,3) follows 
immediately from the equality of the nonmetricity tensor to zero. This result is essentially
a well-known fact, see, for example, the monograph [12]. In our work we note
that, just as in the case of the tetrad formulation of standard general relativity, we
consider the metric spin connection. That is, in the tetrad formulation of conformal
GR, the same spin connection is used as in the tetrad formulation of standard
GR. It follows from this that considering the spin connection with respect to
the SO(1,3) group does not change the essence of the theory, at least not at the
classical level.

We added the above discussion in the middle of Section 2 (the new text is in bold face).

Comment 3:

The real question is whether the symmetry reduction can affect the quantization of the theory. In fact, the quantization of a GL(4) theory is completely different from the quantization of an SO(3,1) theory. The authors should include a detailed discussion about this issue.

Reply 3:

Here the Referee notes correctly that the quantization of the GL(4) theory should be different from the quantization of the SO(1,3) theory. Our work is devoted to the consideration of a special set of variables for the quantization of conformal general relativity associated with spin connection and tetrad formalism. That is, we are studying one of many possible approaches. The question of what differences should exist when quantizing a theory with a symmetry group other than GL(4) is an interesting one. A meaningful study of such a difference is by no means trivial and is the topic of a separate work. It is known that the use of tetrads and spin connections helps to construct a quantum theory of spinor fields in a curved space-time. In this regard, in this work we consider it being interesting to consider a similar approach to gravity. 

We have added a corresponding paragraph with clarifications to Section 2.

Comment 4:

The authors propose the spin connection $\omega^R$ as the underlying variable for quantization. Why? Why not $\omega^L$? This theory contains metric and connection as geometric variables (apart from the dilatonic field). Does the choice of $\omega$ implies that the degrees of freedom of the metric should not be quantized?

Reply 4:

The guiding considerations for quantization in terms of $\omega^R$ variables and not $\omega^L$ ones are related to the fact that (as shown in the Section 2) the expression for the metric differential does not include $\omega^{L}$ variables. This is due to the fact that $\omega^{L}$ is antisymmetric in the indices $a,b$, and $\omega^{R}$ and the rest of the expression turn out to be symmetric (see the end of Section 2). It is natural to consider as quantion variables those ones which describe the dynamics of the metric tensor. Meanwhile, the differential of the metric does not depend on $\omega^{L}$.

If $\omega^{R}$ is known, then the metric tensor and its derivatives can be expressed in terms of $\omega^{R}$. Therefore, as the reviewer correctly noted, that the metric degrees of freedom in this sense are not quantized. Instead, we consider the possibility of quantizing the theory in $\omega^{R}$ variables, which are proposed as the basic variables of quantum gravity.

Comment 5:

The authors use a particular metric for planar waves to analyze the quantization problem and conclude that the corresponding quantum theory is trivial. This is not correct. They can only conclude that the quantization of planar waves cannot be performed in this way. The point is that fixing a gauge and quantizing is not the same as quantizing and fixing the gauge. The authors should discuss in detail their method of quantization from a physical and mathematical point of view because this is certainly not the standard way of looking at the problem of quantizing gravity.

Reply 5:

The reviewer's remark that the results strongly depend on whether the gauge conditions are imposed before or after quantization of the theory is certainly valid. In our work we consider the scenario where we at the first stage we impose gauge conditions and then perform quantization. We propose to proceed by the direct analogy with the quantization of a spinor field in a curved space-time, i.e., first we go to the tangent space and only then we perform quantization.

Round 2

Reviewer 2 Report

Comments and Suggestions for Authors

The authors have responded appropriately to all the questions raised in my report. I recommend publication.

Comments on the Quality of English Language

Only minor improvements are recommended.